

# Oral health status, oral hygiene behaviors, and caries risk assessment of individuals with special needs: a comparative study of Pakistan and Saudi Arabia

Osama Khattak[1], Farooq Ahmad Chaudhary[2], Shahzad Ahmad[3], Muhammad Amber Fareed[4,5], Shazia Iqbal[3], Asma Shakoor[6], Mohammed Nadeem Baig[7], Haifa Ali Almutairi[8], Rakhi Issrani[7] and Azhar Iqbal[1]

[1] Department of Restorative Dentistry, College of Dentistry, Jouf University, Sakaka, Saudi Arabia
[2] School of Dentistry, Shaheed Zulfiqar Ali Bhutto Medical University, Islamabad, Pakistan
[3] Faculty of Medicine and Health Science, The University of Buckingham, Buckingham, United Kingdom
[4] Clinical Sciences Department, College of Dentistry, Ajman University, Ajman, United Arab Emirates
[5] Centre of Medical and Bio-allied Health Sciences Research, Ajman University, Ajman, United Arab Emirates
[6] Community and Preventive Dentistry Department, CMH-Lahore Medical College and Institute of Dentistry, National University of Medical Sciences, Rawalpindi, Pakistan
[7] Department of Preventive Dentistry, College of Dentistry, Jouf University, Sakaka, Saudi Arabia
[8] College of Dentistry, Jouf University, Sakaka, Saudi Arabia

Corresponding authors
Farooq Ahmad Chaudhary,
chaudhary4@hotmail.com
Muhammad Amber Fareed,
m.fareed@ajman.ac.ae

## ABSTRACT

**Background.** Individuals with disabilities often experience greater challenges in managing oral diseases, including dental caries and periodontal conditions, due to functional limitations. This study aims to: (1) assess the oral health status of disabled individuals in Pakistan and Saudi Arabia, (2) evaluate their oral hygiene knowledge and behaviors, and (3) determine their caries risk using the Caries Management by Risk Assessment (CAMBRA) protocol.

**Methods.** A cross-sectional study was conducted on 189 participants aged 13 years and older, including both young people and adults with hearing, visual, or intellectual disabilities from Pakistan and Saudi Arabia between September 2023 and April 2024. The participants were recruited from the Institute of Special Education, Pakistan, and the Saudi Institute of Rehabilitation Medicine, Saudi Arabia. Intraoral examinations and bitewing radiographs assessed oral health, including Decayed, Missing, and Filled Teeth (DMFT) index, Gingival Index, visible plaque, and molar alignment. A self-administered questionnaire gathered sociodemographic data and evaluated oral hygiene knowledge and behaviors. Caries risk was analyzed using the CAMBRA tool. Data were analyzed using descriptive statistics, Chi-square tests, and binary logistic regression.

**Results.** The mean DMFT score was 6.30 (SD = 1.83), with a statistically significant difference between Pakistan and Saudi Arabia ($p = 0.007$). Gingival health was fair to poor in 47% of participants, while 43.4% exhibited bleeding on probing and 34.9% had visible plaque. Class III malocclusion affected approximately 30% of participants in both countries. Tooth brushing frequency showed a significant difference between the two groups ($p = 0.005$). Most participants (76% in Pakistan, 62% in Saudi Arabia) were classified as high caries risk. Deep pits and fissures (69.4%) and frequent snacking (63.8%) were the main risk factors in Pakistan, while frequent snacking (71.6%)
and heavy plaque (60.4%) were prevalent in Saudi Arabia. Saudi participants had a significantly higher likelihood of being in the high-risk group for caries (OR = 1.86, 95% CI [0.95–3.65], $p = 0.04$).

**Conclusion**. The disabled individuals in both countries face significant oral health challenges, with high caries risk and poor oral hygiene practices. Targeted preventive measures and improved dental care access are essential to addressing these disparities.

## INTRODUCTION

People with disabilities, including those with intellectual, emotional, developmental, sensory, or physical impairments, represent a significant and vulnerable group in terms of health outcomes (*Currie & Kahn, 2012*). These conditions, such as Down syndrome, seizure disorders, hearing and vision impairments, and craniofacial anomalies, not only create functional limitations but also complicate the management of secondary health issues, including oral diseases (*Alamri, 2022*; *Chaudhary, 2023*). For instance, the chronicity of dental conditions like dental caries and periodontal diseases often exacerbates the challenges faced by individuals with disabilities, leading to an increased prevalence of oral health problems within this population (*Salles et al., 2012*). Studies have shown that individuals with disabilities often encounter multiple barriers to accessing dental care, which contributes to significant oral health disparities compared to the general population (*Salles et al., 2012*; *Chaudhary, Ahmad & Bashir, 2019*).

This vulnerable group often faces communication difficulties, heightened anxiety, and sensory sensitivities that make traditional dental treatments challenging. Consequently, individuals with disabilities frequently require specialized care and personalized approaches to meet their unique dental needs (*Desai, Messer & Calache, 2001*; *Fazal et al., 2023*). Despite advancements in dental care and treatment options, oral diseases like dental caries remain a major public health issue globally. Dental caries alone affect 60–90% of children and adults worldwide, and individuals with disabilities are often at even higher risk (*Levine & Stillman-Lowe, 2019*).

Earlier studies have emphasized that individuals with disabilities are disproportionately affected by dental caries diseases (*Alamri, 2022*; *Chaudhary, 2023*; *Chaudhary, Ahmad & Bashir, 2019*). Therefore, there is a critical need for effective caries risk assessment models tailored to this group (*Cui et al., 2022*). Assessing caries risk is not only essential for managing and preventing the progression of dental caries but also crucial for providing individualized, cost-effective dental care (*Iqbal et al., 2022*). Traditional approaches to dental care, which do not take individual risk factors into account, may not be suitable for disabled individuals who often require a more targeted approach (*Riley III et al., 2011*).

The Caries Management by Risk Assessment (CAMBRA) has emerged as an evidence-based protocol aimed at identifying individuals at high risk for dental caries at the earliest

possible stage (*Iqbal et al., 2022*; *Iqbal et al., 2024*). This risk-based model enables dental professionals to evaluate disease indicators, risk factors, and protective factors to classify individuals into low, moderate, high risk categories. CAMBRA provides a systematic and minimally invasive approach to managing dental caries by focusing on prevention and addressing the root causes of disease rather than simply treating the symptoms (*Iqbal et al., 2024*). This approach is particularly beneficial for individuals with disabilities, whose unique conditions may predispose them to increased caries risk (*Maheswari et al., 2015*).

While CAMBRA has been widely adopted in general populations, its application in individuals with disabilities remains underexplored, especially in developing countries like Pakistan and also in developed country like Saudi Arabia. The current literature largely focuses on caries risk in general pediatric or adult populations without considering the specific challenges faced by those with disabilities (*Iqbal et al., 2022*; *Ng et al., 2024*). Furthermore, studies exploring oral health status and caries risk in disabled individuals have often been small-scale or single-center with single specific disablity, limiting the generalizability of their findings (*Marshall, Sheller & Mancl, 2010*; *Alkhabuli et al., 2019*). There is also a notable lack of comparative studies examining these issues across different countries with economic status, varying healthcare systems and socio-cultural contexts. For example, Saudi Arabia offers universal healthcare with specialized services for individuals with disabilities, while Pakistan faces challenges related to healthcare access and funding (*Nair et al., 2024*; *Khan et al., 2023*). Similarly, differences in cultural practices and health literacy further shape oral health behaviors and outcomes, making cross-country comparisons essential for understanding diverse influences on caries risk in disabled populations. The comparison between the two countries is valuable because it allows us to examine how differing levels of healthcare infrastructure, access to specialized care, and socio-cultural factors influence the oral health outcomes of disabled individuals (*Lima et al., 2024*).

Thus, the objectives of this study are threefold: (1) to explore the oral health status of disabled individuals in Pakistan and Saudi Arabia, (2) to assess their oral hygiene knowledge and Behaviors, and (3) to assess their caries risk using the CAMBRA protocol. By addressing these objectives, this study aims to fill the existing gaps in the literature regarding the oral health needs of disabled individuals in these two culturally distinct countries. This research will provide valuable insights into the specific challenges faced by disabled populations in accessing oral healthcare and offer evidence-based recommendations for improving preventive dental care for this vulnerable group.

## MATERIALS AND METHODS

This cross-sectional study was conducted in Pakistan and Saudi Arabia between September 2023 and April 2024, employing a non-probability convenience sampling technique. The study focused on disabled individuals, with participants recruited from the Institute of Special Education and Bahria College for Special Needs in Islamabad, Pakistan, and the Saudi Institute of Rehabilitation Medicine in Jouf, Saudi Arabia. The study population included teenagers, adults and young aged 13 years and older with hearing, visual, and

intellectual disabilities. The age cut-off of 13 years was selected to align with a developmental stage where adolescents typically transition to adult dental care services, allowing for a more uniform assessment of oral health behaviors and risk factors. Participants with other types of disabilities (autism, cerebral palsy, or other neurological and developmental disorders) and those whose parents or guardians declined to consent were excluded. Efforts were made to minimize selection bias by clearly defining the inclusion criteria and recruiting from well-established institutions specializing in special education and rehabilitation. The study focused on individuals with hearing, visual, and intellectual disabilities, as these conditions are commonly associated with oral health disparities and present a range of challenges for dental care. Other disability types, such as autism and cerebral palsy, were excluded due to the unique methodological and care-related considerations they present. These conditions often require more specialized examination procedures and could introduce additional variability that might compromise the comparability of findings. Moreover, individuals with autism or cerebral palsy may have distinct oral health challenges that would require separate, in-depth studies to assess properly.

The sample size for each country was calculated using data from reference studies. In Pakistan, the sample size was derived from a previous study on oral hygiene status among individuals with special needs in Karachi (*Azfar et al., 2018*), using a prevalence of 50%, a margin of error of 9%, and a 95% confidence level, resulting in 109 participants. For Saudi Arabia, the sample size was based on a study of hearing-impaired participants (*Al-Qahtani et al., 2017*), with a prevalence of 95% for dental caries, a margin of error of 4.65%, and a 95% confidence level, yielding 85 participants. The sample size for Saudi Arabia was initially estimated based on a study targeting a specific type of disability, with margins of error selected based on population characteristics, differences in healthcare infrastructure, disability-specific care, access to specialized care, socio-cultural factors and prior studies which can influence oral health behaviors and outcomes. In Saudi Arabia, the margin of error was based on a study focused on a specific disability type. During recruitment, it was adjusted to include a broader range of disabilities. In Pakistan, where disability-specific data was limited, the margin of error was set to account for greater variability. These differences may affect variability between the two populations. However, the margins of error were designed to ensure sufficient precision for cross-country comparisons while remaining robust for analyzing oral health differences in each context.

Ethical approval for this study was obtained from the Ethical Review Board of the School of Dentistry, Shaheed Zulfiqar Ali Bhutto Medical University, SZABMU (Ref. No. SOD/ERB/2023/156). Written informed consent was obtained from participants who could comprehend the study's purpose. For minors and individuals with cognitive impairments, consent was obtained from parents or guardians.

All participants underwent intraoral examinations and bitewing radiographs to assess their oral health, particularly for the detection of caries. A self-administered questionnaire was also completed. Trained investigators conducted all oral examinations under standardized conditions, using appropriate lighting and equipment in a reclined chair setting. Before initiating the study, all examiners underwent a three-day calibration workshop led by an experienced dentist in their respective countries and final joint

session conducted online to ensure consistency, uniform understanding and application of assessment criteria across all examiners. The training included theoretical sessions on the CAMBRA protocol, oral health indices (DMFT, Gingival Index, and Visible Plaque Index), and practical exercises on patient evaluation and scoring. To evaluate inter-examiner reliability, five participants were examined independently by all study examiners, and their scores were compared using Cohen's kappa statistics. A kappa value of 0.85 was achieved, indicating strong agreement. Intra-examiner reliability was assessed by re-examining the same five participants within one week, yielding a kappa value of 0.88. Both reliability tests exceeded the acceptable threshold of 0.75, ensuring consistency in data collection. These procedures helped minimize measurement bias and ensured data quality and reproducibility. Examination tools included a mouth mirror, explorer, and periodontal probe.

The Decayed, Missing, and Filled Teeth (DMFT) index was used to measure the participants' caries experience (*World Health Organization, 2013*). The total number of decayed, missing, and filled teeth was recorded for each participant, with a higher DMFT score indicating a greater burden of caries. The Loe & Silness Gingival Index was used to assess gingival health (*World Health Organization, 2013*). Six index teeth (molars and incisors) were examined, with each tooth's buccal, mesial, lingual, and distal surfaces scored from 0 (healthy) to 3 (severe inflammation). Participants were categorized based on their scores into four groups: Excellent (score 0), Good (0.1–1.0), Fair (1.1–2.0), and Poor (2.1–3.1). Oral hygiene was assessed by identifying visible plaque on the maxillary incisors using a periodontal probe (*World Health Organization, 2013*). Plaque was recorded as either "Yes" (visible plaque present) or "No" (no visible plaque), offering a simple yet reliable assessment of oral hygiene status. Angle's Classification of Malocclusion was used to evaluate molar alignment, categorizing participants into Class I (normal), Class II (distocclusion), or Class III (mesiocclusion) (*Dewey, 1915*). The relationship between the mesiobuccal cusp of the maxillary first molar and the buccal groove of the mandibular first molar was assessed.

After the intraoral examination, participants completed a 17-item self-administered questionnaire. For those with visual impairments or comprehension difficulties, questions and response options were read aloud by the research team or school staff. The questionnaire, was adapted from *Tadin et al. (2022)*. To ensure its relevance and appropriateness for our study objectives and target population, we conducted a thorough review and validation process in both countries. An expert panel consisting of three members—an Assistant Professors of Operative Dentistry, an Assistant Professor of Epidemiology, and a specialist in special care dentistry in Pakistan and almost similar panel in Saudi Arabia reviewed the questionnaire items. Their feedback was incorporated into the final version of the questionnaire. The questionnaire was divided into two sections: (1) Sociodemographic information: Including age, gender, parental education, occupation, and length of employment. (2) Oral hygiene knowledge and behaviors: Assessing knowledge of tooth brushing practices, oral health's impact on general health, the effects of carbonated drinks and sugary foods, and the sources of oral health information.

It also evaluated behaviors like brushing frequency, duration, toothpaste type, dental visits, and independence in tooth brushing.

Caries risk was assessed using a customized CAMBRA tool, evaluating eight risk factors, protective factors, and disease indicators (*Iqbal et al., 2024*). The CAMBRA protocol has been validated and successfully implemented in both Pakistan and Saudi Arabia in previous studies (*Iqbal et al., 2022*; *Iqbal et al., 2024*). Participants were classified into low, moderate, or high-risk groups based on factors such as plaque presence, fluoride use, dental care history, and salivary function. Certain components of the CAMBRA protocol, such as bacterial counts, were excluded from this study due to financial and logistical constraints. Performing bacterial counts requires specialized laboratory facilities, which were not feasible within the context of this study. Additionally, the decision was made to focus on more practical and accessible indicators, including plaque presence, fluoride use, and dental care history, which are reliable and cost-effective means of assessing caries risk.

Descriptive statistics were computed to summarize the data, with frequency distributions for categorical variables. Chi-square tests were used to assess the association between oral health indicators (chi-square test is well-suited for examining the association between categorical variables and assessing whether the distribution of categories differs between groups) and sociodemographic factors. Binary logistic regression analysis was employed to explore relationships between caries risk categories and sociodemographic variables. Inter-examiner reliability was assessed using Cohen's kappa statistic, with values $\geq 0.75$ considered acceptable. Kappa values were calculated after the calibration sessions to ensure consistency in the assessment of oral health indicators. Effect sizes were calculated using Cohen's d for mean comparisons and for categorical data, Cramer's V was used. All statistical analyses were performed using IBM SPSS (version 25.0), with a significance level set at $p < 0.05$.

# RESULTS

A total of 189 participants with disabilities were included in the study, with 108 participants from Pakistan and 81 from Saudi Arabia. The majority of the participants were male (61.9%) and aged between 13–16 years (66.1%). The distribution of participants based on disability was as follows: Intellectual disability/Down syndrome (48.1%), hearing loss (38.1%), and blindness (13.8%). Most of the participants' parents had a school-level education (70.4%) and had more than 11 years of working experience (71.4%) and only 13% works in government sector (Table 1). The caries risk assessment indicated that the majority of participants in both countries were classified as having a high risk of caries (69.8%), with 75.9% in Pakistan and 61.7% in Saudi Arabia (Fig. 1).

The mean DMFT score for the participants was 6.30 (SD = 1.83). A statistically significant difference was found between the DMFT scores of participants from Pakistan (6.61, SD = 1.81) and Saudi Arabia (5.89, SD = 1.78) ($p = 0.007$) (Fig. 2). The effect size, as measured by Cohen's d, was 0.40, indicating a small to medium effect size. This suggests that the difference in oral health status between the two groups is notable but not large enough to drive immediate clinical decisions or interventions but highlights a

| Characteristics | Pakistan N (%) 108 (57.1) | Saudi Arabia N (%) 81 (42.9) | Total N (%) 189 |
|---|---|---|---|
| **Age** | | | |
| 13–16 | 70 (64.8) | 55 (67.9) | 125 (66.1) |
| 17–26 | 32 (29.6) | 21 (25.9) | 53 (28.0) |
| >27 | 6 (5.6) | 5 (6.2) | 11 (5.8) |
| **Gender** | | | |
| Male | 68 (63.0) | 49 (60.5) | 117 (61.9) |
| Female | 40 (37.0) | 32 (39.5) | 72 (38.1) |
| **Education of parents** | | | |
| No-formal | 2 (1.9) | 2 (2.5) | 4 (2.1) |
| School | 74 (68.5) | 59 (72.8) | 133 (70.4) |
| College | 20 (18.5) | 13 (16.0) | 33 (17.5) |
| University | 12 (11.1) | 7 (8.6) | 19 (10.1) |
| **Occupation of parents** | | | |
| Labor | 6 (5.6) | 3 (3.7) | 9 (4.8) |
| Government | 14 (13.0) | 9 (11.1) | 23 (12.2) |
| Professional | 36 (33.3) | 27 (33.3) | 63 (33.3) |
| Other | 52 (48.1) | 42 (51.9) | 94 (49.7) |
| **Length of employment (years)** | | | |
| <5 | 19 (17.6) | 8 (9.9) | 27 (14.3) |
| 5–10 | 17 (15.7) | 10 (12.3) | 27 (14.3) |
| 11–15 | 44 (40.7) | 38 (46.9) | 82 (43.4) |
| >16 | 28 (25.9) | 25 (30.9) | 53 (28.0) |
| **Disability of individuals** | | | |
| Hearing loss | 36 (33.3) | 36 (44.4) | 72 (38.1) |
| Blind | 16 (14.8) | 10 (12.3) | 26 (13.8) |
| Intellectual disability (ID)/Down Syndrome | 56 (51.9) | 35 (43.2) | 91 (48.1) |

**Table 1** Sociodemographic characteristics of the participants.

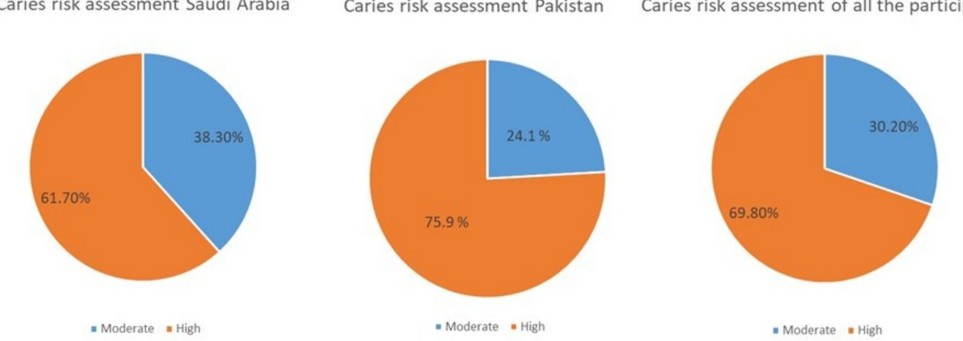

**Figure 1** Distribution of study population according to caries risk assessment.

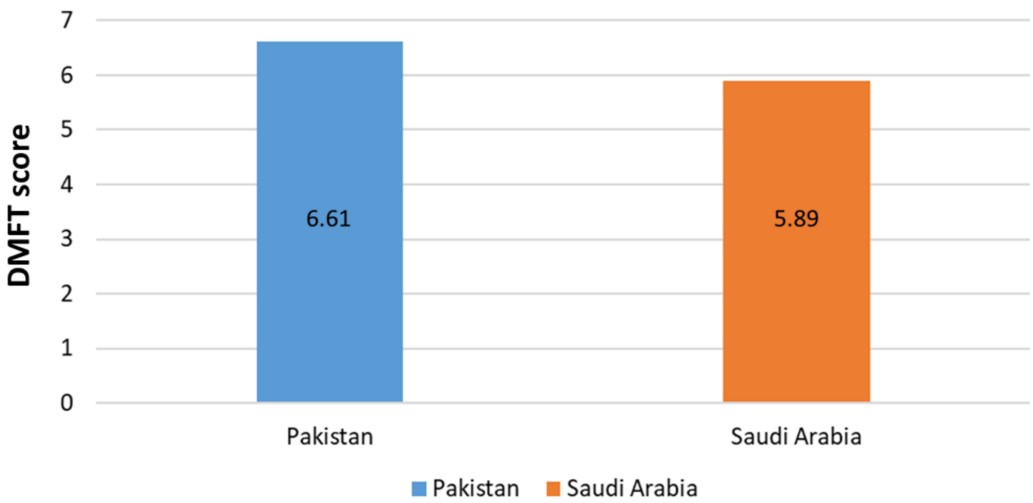

**Figure 2 Comparison of the DMFT scores between two countries.**

meaningful trend that warrants further investigation. A small to medium effect suggests that while disparities exist, large-scale policy changes or interventions may require additional supporting evidence. However, targeted preventive measures could still be beneficial.

A total of 47% of participants exhibited fair to poor gingival health, with 43.4% showing bleeding on probing, and 34.9% having visible plaque. Molar alignment was primarily in Class I angle classification for 55.6% of Pakistani participants and 60.5% of Saudi participants, though approximately 30% of participants in both countries had Class III malocclusion (Table 2).

The results for oral health behaviors and knowledge are summarized in Table 3. A statistically significant difference was observed in the frequency of tooth brushing between participants from Pakistan and Saudi Arabia ($p = 0.005$). Among Saudi participants, 34.6% reported brushing their teeth twice daily, compared to only 13.0% of participants from Pakistan. The effect size, measured by Cramer's V (0.151), indicates that while the difference is statistically significant, the magnitude of this difference is small. Overall, 97% of participants reported using a toothbrush to clean their teeth, with 58% brushing in the morning.

Despite these habits, 57% of participants were unaware of the type of toothpaste they should use, and 54% had not visited a dentist in the past year. Furthermore, 37% of participants were unable to brush their teeth independently, highlighting a critical barrier to effective oral hygiene in this population.

In terms of oral health knowledge, 59% of participants believed that oral health affects general health, and 70.4% recognized the importance of proper tooth brushing in maintaining oral hygiene. A higher proportion (77.2%) understood that sugary and sticky foods could damage teeth, but awareness of the harmful effects of carbonated drinks was lower, with only 50.8% acknowledging this.

**Table 2** The summary of the DMFT, Gingival Index, bleeding gums, visible plaque and molar classification of the participants in Pakistan and Saudi Arabia.

| Oral Health Status | Pakistan N (%) | Saudi Arabia N (%) | Total N (%) | P-value |
|---|---|---|---|---|
| Gingival Index | | | | |
| Excellent | 9 (8.3) | 11 (13.6) | 20 (10.6) | 0.548 |
| Good | 46 (42.6) | 35 (43.2) | 81 (42.9) | |
| Fair | 36 (33.3) | 21 (25.9) | 57 (30.2) | |
| Poor | 17 (15.7) | 14 (17.3) | 31 (16.4) | |
| Bleeding | | | | |
| Yes | 45 (41.7) | 37 (45.7) | 82 (43.4) | 0.582 |
| No | 63 (58.3) | 44 (54.3) | 107 (56.6) | |
| Visible plaque | | | | |
| Yes | 37 (34.3) | 29 (35.8) | 66 (34.9) | 0.826 |
| No | 71 (65.7) | 52 (64.2) | 123 (65.1) | |
| Molar class | | | | |
| Class I | 60 (55.6) | 49 (60.5) | 109 (57.7) | 0.760 |
| Class II | 15 (13.9) | 9 (11.1) | 24 (12.7) | |
| Class III | 33 (30.6) | 23 (28.4) | 56 (29.6) | |

Parents were identified as the primary source of information on maintaining oral health for 63% of participants. Additionally, 57.1% of participants reported that their primary caregiver or school teacher educated them about oral health and its maintenance.

Most participants in both countries had one or more disease indicators, with the most common being visible cavities (Pakistan = 98.1%, Saudi Arabia = 98.7%) and radiographic approximal enamel lesions (Pakistan = 88.8%, Saudi Arabia = 83.9%). The most common risk factor in Pakistan was deep pits and fissures (69.4%), followed by frequent snacking (63.8%). In Saudi Arabia, the most common risk factor was frequent snacking (71.6%), followed by visible heavy plaque (60.4%). The most frequently reported protective factor was the use of fluoridated toothpaste at least once daily (Pakistan = 68.5%, Saudi Arabia = 77.7%) (Table 4).

In the logistic regression analysis, adjusting for country and other sociodemographic factors, being from Saudi Arabia was significantly associated with a higher likelihood of participants being in the high-risk group for caries. Disabled participants from Saudi Arabia were 1.86 times more likely to be in the high-risk group compared to those from Pakistan (OR = 1.86, 95% CI [0.95–3.65], $p$ = 0.04) (Table 5). Other all sociodemographic variables were not significantly associated with caries risk categories.

## DISCUSSION

The findings of this study highlight significant oral health challenges among disabled individuals in Pakistan and Saudi Arabia, with a particular focus on caries risk, oral hygiene behaviors, and gingival health. This study contributes to the growing body of literature indicating that individuals with disabilities face disproportionate risks for oral diseases due to a combination of biological, behavioral, and systemic factors.

**Table 3** The summary of oral health behaviors and knowledge among participants of Pakistan and Saudi Arabia.

| Oral hygiene practice | Total N (%) | Pakistan N (%) | Saudi Arabia N (%) | P-Value |
|---|---|---|---|---|
| How many times do you brush your teeth? | | | | 0.005 |
| Once a day | 127 (67.2) | 82 (75.9) | 45 (55.6) | |
| Twice | 42 (22.2) | 14 (13.0) | 28 (34.6) | |
| Thrice | 11 (5.8) | 6 (5.6) | 5 (6.2) | |
| Do not brush | 9 (4.8) | 6 (5.6) | 3 (3.7) | |
| Do you know how should teeth be cleaned? | | | | 0.392 |
| Finger | 5 (2.6) | 4 (3.7) | 1 (1.2) | |
| Stick | 0 | 0 | 0 | |
| Tooth-brush | 183 (96.8) | 103 (95.4) | 80 (98.8) | |
| Charcoal | 1 (0.5) | 1 (0.9) | 0 | |
| How long should you clean your teeth? | | | | 0.485 |
| Less than a minute | 20 (10.6) | 14 (13.0) | 6 (7.4) | |
| 2 min | 103 (54.5) | 55 (50.9) | 48 (59.3) | |
| 5 min | 36 (19.0) | 20 (18.5) | 16 (19.8) | |
| I don't know | 30 (15.9) | 19 (17.6) | 11 (13.6) | |
| Are you able to brush independently? | | | | 1.00 |
| Yes | 119 (63.0) | 68 (63.0) | 51 (63.0) | |
| No | 70 (37.0) | 40 (37.0) | 30 (37.0) | |
| **Oral health knowledge** | | | | |
| Do you think oral health has an effect on general health? | | | | 0.468 |
| Yes | 111 (58.7) | 61 (56.5) | 50 (61.7) | |
| No | 78 (41.3) | 47 (43.5) | 31 (38.3) | |
| Do you think proper tooth brushing helps in maintaining oral hygiene? | | | | 0.748 |
| Yes | 133 (70.4) | 75 (69.4) | 58 (71.6) | |
| No | 56 (29.6) | 33 (30.6) | 23 (28.4) | |
| Do you think carbonated drinks have adverse effects on teeth? | | | | 0.401 |
| Yes | 93 (49.2) | 56 (51.9) | 37 (45.7) | |
| No | 96 (50.8) | 52 (48.1) | 44 (54.3) | |
| Do you think sugary/sticky food items can damage teeth? | | | | 0.394 |
| Yes | 146 (77.2) | 81 (75.0) | 65 (80.2) | |
| No | 43 (22.8) | 27 (25.0) | 16 (19.8) | |
| Which type of toothpaste should be used? | | | | 0.175 |
| Fluoride containing | 77 (40.7) | 38 (35.2) | 39 (48.1) | |
| Without fluoride | 4 (2.1) | 3 (2.8) | 1 (1.2) | |
| I don't know | 108 (57.1) | 67 (62.0) | 41 (50.6) | |
| What do you know is the ideal time for brushing your teeth? | | | | 0.860 |
| In the morning | 111 (58.7) | 64 (59.3) | 47 (58.0) | |
| At night | 51 (27.0) | 27 (25.0) | 24 (29.6) | |
| After every meal | 19 (10.1) | 12 (11.1) | 7 (8.6) | |
| All of the above | 8 (10.1) | 5 (4.6) | 3 (3.7) | |

**Table 3** (*continued*)

| Oral hygiene practice | Total N (%) | Pakistan N (%) | Saudi Arabia N (%) | P-Value |
|---|---|---|---|---|
| **Dental care access and education** | | | | |
| Did your primary care giver or school-teacher educate you regarding oral health and maintenance? | | | | 0.161 |
| Yes | 108 (57.1) | 57 (52.8) | 51 (63.0) | |
| No | 81 (42.9) | 51 (47.2) | 30 (37.0) | |
| Have you been to a dentist during the past years? | | | | 0.613 |
| Yes | 87 (46.0) | 48 (44.4) | 39 (48.1) | |
| No | 102 (54.0) | 60 (55.6) | 42 (51.9) | |
| From where did you get information on how to keep your mouth clean? | | | | 0.128 |
| Parents | 119 (63.0) | 73 (67.6) | 46 (56.8) | |
| Teacher | 70 (37.0) | 35 (32.4) | 35 (43.2) | |

**Table 4  Distribution according to disease indicators, risk, and protective factors among participants of Pakistan and Saudi Arabia.**

| Disease indicators | Pakistan N (%) | Saudi Arabia N (%) |
|---|---|---|
| Visible cavities or radiographic penetration of the dentin | 106 (98.1) | 80 (98.7) |
| Radiographic approximal enamel lesions (not in dentin) | 96 (88.8) | 68 (83.9) |
| White spots on smooth surfaces | 67 (62.0) | 59 (72.8) |
| Restorations in last three years | 52 (48.1) | 61 (75.3) |
| **Risk factors** | | |
| Visible heavy plaque on teeth | 37 (34.2) | 49 (60.4) |
| Frequent snack (>3 _ daily between meals) | 69 (63.8) | 58 (71.6) |
| Deep pits and fissures | 75 (69.4) | 44 (54.3) |
| Recreational drug use | 2 (1.8) | 0 |
| Inadequate saliva flow by observation | 56 (51.8) | 35 (43.2) |
| Saliva reducing factors (medications/radiation/systemic) | 61 (56.4) | 37 (45.6) |
| Exposed roots | 31 (28.7) | 20 (24.6) |
| Orthodontic appliances | 23 (21.2) | 38 (46.9) |
| **Protective factors** | | |
| Home/work/school is a fluoridated community | 26 (24.0) | 54 (66.6) |
| Fluoride toothpaste at least once daily | 74 (68.5) | 63 (77.7) |
| Fluoride toothpaste at least 2 _ daily | 13 (12.0) | 30 (37.0) |
| Fluoride mouth rinse (0.05% NaF) daily | 15 (13.8) | 25 (30.8) |
| Fluoride varnish in last six months | 5 (4.6) | 12 (14.8) |
| Chlorhexidine prescribed/used one week each of last six months | 3 (2.7) | 7 (8.6) |
| Xylitol gum/lozenges 4 _ daily last six months | 4 (3.7) | 6 (7.4) |
| Calcium and phosphate paste during last six months | 3 (2.7) | 4 (4.9) |

**Table 5  Binary logistic regression of the analysis to explore relationships between caries risk categories and sociodemographic variables.**

| Predictor variable | Coefficient ($\beta$) | Odds ratio (OR) | 95% CI for OR | *P*-value |
|---|---|---|---|---|
| Age (ref.: 13–16 years) | | | | |
| 17–26 | −0.59 | 0.55 | 0.12, 2.45 | 0.43 |
| >27 | −0.11 | 0.89 | 0.17, 4.59 | 0.89 |
| Gender (ref.: Male) | | | | |
| Female | 0.47 | 1.60 | 0.80, 3.22 | 0.18 |
| Education of parents (ref.: No formal) | | | | |
| School | −21.9 | 0.00 | 0 | 0.99 |
| College | −0.26 | 0.76 | 0.18, 3.17 | 0.71 |
| University | −0.17 | 0.84 | 0.18, 3.79 | 0.82 |
| Occupation of parents (ref.: Labor) | | | | |
| Government | 19.8 | 419.7 | 0 | 0.99 |
| Professional | 0.21 | 1.24 | 0.42, 3.61 | 0.69 |
| Other | −0.008 | 0.99 | 0.40, 2.43 | 0.98 |
| Length of Employment (yrs) (ref.: <5) | | | | |
| 5–10 | −0.49 | 0.61 | 0.17, 2.09 | 0.43 |
| 11–15 | −0.73 | 0.48 | 0.14, 1.55 | 0.22 |
| >16 | −0.20 | 0.81 | 0.32, 2.04 | 0.66 |
| Disability of individuals children (ref.: Hearing loss) | | | | |
| Blind | −0.6 | 0.52 | 0.24, 1.11 | 0.09 |
| Intellectual Disability (ID)/Down Syndrome | 0.90 | 2.47 | 0.58, 10.4 | 0.21 |
| Country (ref.: Pakistan) | | | | |
| Saudi Arabia | 0.67 | 1.86 | 0.95, 3.65 | 0.04 |

The high prevalence of dental caries observed in both populations, with 75.9% of participants in Pakistan and 61.7% in Saudi Arabia classified as high-risk, is consistent with findings from previous studies on individuals with disabilities worldwide. For instance, a study by *Shyama et al. (2001)* in Kuwait reported similarly elevated caries rates in disabled populations, with 80% classified as high caries risk. This aligns with broader research indicating that individuals with disabilities are at greater risk of dental caries compared to their non-disabled counterparts, largely due to challenges in maintaining oral hygiene and accessing dental care.

The mean DMFT score of 6.30 observed in this study is comparable to findings from other regions. For example, *Al Habashneh et al. (2012)* reported DMFT scores ranging from 5.5 to 7.1 in disabled populations in Jordan. The statistically significant difference in DMFT scores between Pakistan and Saudi Arabia ($p = 0.007$) may reflect disparities in public health policies, dietary practices, and access to fluoridated water between the two countries.

Furthermore, country-specific caries risk factors identified in this study underscore the importance of tailored preventive strategies. In Pakistan, the high prevalence of deep pits and fissures (69.4%) was a prominent risk factor, while frequent snacking was more common in Saudi Arabia (71.6%). These findings highlight the need for targeted

interventions that address the unique risk profiles of each population to effectively reduce caries prevalence and improve oral health outcomes.

The poor oral hygiene practices among disabled individuals in both countries are concerning, with 47% of participants exhibiting fair to poor gingival health, 43.4% presenting with bleeding on probing, and 34.9% showing visible plaque. These findings are consistent with previous studies that have highlighted the challenges faced by disabled individuals in maintaining good oral hygiene. Studies in Brazil and India also reported poor oral hygiene and gingival health among individuals with intellectual and developmental disabilities (*Braúna et al., 2016*; *Makkar et al., 2019*; *Mehta et al., 2015*). The significant difference in tooth brushing frequency between Saudi and Pakistani participants ($p = 0.005$) suggests that cultural or educational factors may play a role in oral hygiene behaviors. Saudi participants, who brushed more frequently, may benefit from more accessible health education campaigns or greater parental involvement. However, despite higher brushing frequency, the caries risk remained high, emphasizing that brushing alone is insufficient without addressing other factors like diet, fluoride use, and regular dental visits.

The lack of knowledge regarding toothpaste type (57%) and irregular dental visits (54% had not seen a dentist in the past year) further complicates efforts to improve oral health in this population. Similar knowledge gaps have been reported in other studies on disabled individuals, particularly in developing countries where oral health education is often inadequate (*Al-Sufyani et al., 2014*; *Chu et al., 2008*; *Anwar et al., 2022*). This highlights the critical need for improved oral health education programs that are tailored to the needs of disabled individuals and their caregivers.

Our findings reveal an interesting disconnect between self-reported oral hygiene behaviors and clinical outcomes. Despite Saudi participants reporting significantly higher twice-daily brushing rates (34.6% *vs.* 13.0% in Pakistan), their rates of visible plaque (35.8% *vs.* 34.3%) and gingival inflammation remained comparable. This discrepancy suggests several possible explanations: (1) Brushing quality may be suboptimal, as frequency alone does not guarantee effective plaque removal, particularly among individuals with disabilities. (2) Self-reported behaviors may be subject to reporting bias, with participants overestimating their oral hygiene practices. (3) Other factors such as diet, fluoride exposure, and genetic predisposition may play a more significant role in determining clinical outcomes than brushing frequency alone. This observation underscores the need for comprehensive oral health education that goes beyond promoting brushing frequency. Future interventions should emphasize proper brushing techniques, duration, and caregiver involvement, particularly for individuals with disabilities who may face additional challenges in maintaining effective oral hygiene.

The use of the CAMBRA protocol in this study offers valuable insights into caries risk management for disabled populations. CAMBRA's focus on both risk and protective factors enables a more nuanced understanding of caries risk in this vulnerable group. For example, the use of fluoridated toothpaste, a key protective factor, was higher in Saudi participants (77.7%) compared to Pakistani participants (68.5%), which may explain the slightly lower caries risk in the Saudi cohort. However, the high prevalence of disease indicators, such as visible cavities (98.1% in Pakistan and 98.7% in Saudi Arabia), underscores the need

for earlier interventions and more aggressive preventive measures. The logistic regression analysis revealed that being from Saudi Arabia was significantly associated with higher odds of being in the high-risk caries group compared to being from Pakistan (OR = 1.86, $p = 0.04$). This finding could be influenced by a combination of dietary habits, cultural differences, and healthcare access. Saudi Arabia's higher consumption of sugary snacks and beverages, as identified in previous studies, could be a contributing factor (*Alsubaie, 2017*; *Chaudhary et al., 2024*; *Khattak et al., 2022*).

An important finding requiring careful interpretation is the seeming contradiction between lower DMFT scores in Saudi participants and their higher odds of being classified in the high-risk caries group. This paradox can be explained by several factors. First, the DMFT index measures past caries experience (a lagging indicator), while the CAMBRA protocol assesses current and future risk based on present behaviors and clinical indicators. The higher prevalence of risk factors among Saudi participants, such as frequent snacking (71.6%) and visible heavy plaque (60.4%), suggests that while their current caries burden is lower, they may be at greater risk for future caries development. Second, differences in healthcare access and treatment approaches may contribute to this pattern. Saudi Arabia's better access to dental care may have resulted in earlier detection and treatment of caries, leading to lower DMFT scores. However, this does not necessarily mean better oral health behaviors, as untreated risk factors may continue to increase future caries risk. This finding underscores the importance of considering both past disease experience and current risk factors when designing preventive oral health strategies, rather than relying solely on traditional caries experience measures like DMFT.

Socioeconomic disparities between Pakistan and Saudi Arabia could have impacted the observed differences in oral health status, behaviors, and caries risk. Saudi Arabia, as a high-income country, has greater access to resources for oral health promotion, preventive services, and treatment along with a more established and better-funded healthcare system that provides free or subsidized dental care for its citizens. In contrast, Pakistan, being a lower-middle-income country, faces economic constraints that limit healthcare access and availability of oral health education programs. Dental care is largely privatized, which poses financial barriers for many families, limiting their ability to seek routine dental care. These socioeconomic differences likely contribute to variations in DMFT scores, oral hygiene behaviors, and caries risk. Future studies could further quantify the role of socioeconomic variables by including household income, parental education, and employment type as direct measures.

Cultural practices and norms play a significant role in shaping oral health behaviors, which in turn affect clinical oral health outcomes. For instance, in Saudi Arabia, the widespread use of miswak has been associated with reduced plaque accumulation and improved gingival health. Miswak's mechanical cleansing action and antimicrobial properties may contribute to lower gingival inflammation, which aligns with the relatively better gingival index scores observed in this population (*Nordin et al., 2020*). Conversely, in Pakistan, oral hygiene behaviors may be influenced by lower health literacy and dietary patterns high in cariogenic foods, potentially contributing to higher DMFT scores and plaque levels (*Chaudhary, Ahmad & Bashir, 2019*). Limited access to fluoridated water and

oral health services in some areas may further exacerbate dental caries risk. Additionally, the role of caregivers in maintaining oral hygiene among individuals with disabilities differs culturally, which may contribute to variations in oral hygiene index (OHI-S) scores. In settings where caregivers actively assist with brushing and oral care, better plaque control and gingival health may be observed, while in environments where self-care is emphasized despite physical limitations, oral hygiene outcomes may be compromised. Understanding these behavioral influences on clinical measures is crucial for interpreting cross-country differences in oral health status and designing culturally appropriate public health interventions. Future research should quantitatively assess the direct impact of these behaviors on clinical outcomes to strengthen causal inferences.

This study contributes to the existing body of literature on oral health disparities by providing one of the first cross-country comparisons of oral health status, hygiene behaviors, and caries risk among individuals with disabilities in Pakistan and Saudi Arabia using the CAMBRA protocol. The findings highlight critical gaps in oral health outcomes and behaviors among this vulnerable population and emphasize the need for culturally tailored interventions in both countries. Previous research has often overlooked the oral health needs of individuals with disabilities, particularly in developing and middle-income countries. The study offers valuable insights into how socioeconomic, cultural, and healthcare system differences influence oral health outcomes. The results of this study have important implications for dental practitioners and public health policymakers. First, the findings of this study suggest that dental professionals in both Pakistan and Saudi Arabia may face challenges in effectively addressing the oral health needs of disabled individuals. While the study did not directly assess training gaps, it underscores the importance of improving education and care models for this population. Based on these observations and supported by existing literature, there is a clear recommendation for specialized training for dental professionals to better manage the oral health needs of individuals with disabilities (*Alamri, 2022*; *Lim et al., 2022*). Second, public health policies should prioritize access to preventive dental care for disabled individuals, including regular dental check-ups, fluoride treatments, and dietary counseling. The disparities in oral health outcomes between Pakistan and Saudi Arabia indicate that country-specific interventions are needed (*Chaudhary et al., 2024*). For example, in Saudi Arabia, where soft drink consumption is among the highest globally (*Aljaadi et al., 2023*; *Siddiqui et al., 2025*), policies should focus on taxation of sugary beverages, stricter advertising regulations, and public awareness campaigns emphasizing the link between sugar consumption and oral diseases. Additionally, expanding preventive oral health programs in primary healthcare centers can help mitigate early-stage dental diseases. In Pakistan, where access to fluoridated water is limited, policy efforts should prioritize water fluoridation initiatives in urban areas and fluoride supplementation programs in non-fluoridated regions. Given lower levels of oral health literacy, nationwide school-based oral health programs tailored for children with disabilities should be implemented, focusing on proper brushing techniques and caries prevention. Furthermore, training primary healthcare workers to provide basic preventive dental care can enhance service accessibility, particularly in underserved areas. Both countries should integrate community-based fluoride varnish applications and caregiver

training programs into existing healthcare models to reduce caries risk in high-risk groups. Strengthening insurance coverage for dental care, especially for vulnerable populations, can further enhance oral health equity.

This study has several limitations that warrant consideration. The non-probability convenience sampling method limits the generalizability of the findings and may have introduced selection bias, however, it was the most feasible method given the challenges of recruiting individuals with disabilities. The results are most applicable to similar populations within the two countries, but the influence of different healthcare systems, cultural factors, and socioeconomic conditions may limit their broader applicability. Further studies are needed to validate these findings in other regions and among other populations with disabilities. Future studies should aim for a more diverse and randomly selected sample to improve the generalizability of findings to the broader disabled population. As a cross-sectional study, this design limits our ability to make causal inferences about the relationships between oral health behaviors and outcomes. Longitudinal studies are needed to examine the directionality of these associations and to explore potential causal pathways. Subgroup analyses by disability type were not conducted due to small sample sizes, which would reduce statistical power and increase the risk of Type II errors. Additionally, our study's primary focus was on cross-country comparisons rather than intra-group differences. Given the exploratory nature of this research, future studies with larger, disability-specific samples are needed to examine variations across different disability categories. The separate sample size estimations and recruitment strategies for Pakistan and Saudi Arabia allowed for detailed insights into each country's oral health status among disabled populations. However, this approach does not facilitate robust cross-country comparisons. This study did not perform power calculations specifically for cross-country comparisons, which limits the statistical power for detecting significant differences and the differences in sample sizes and margins of error between the countries further constrain the ability to draw definitive conclusions. Future research should consider power calculations tailored to cross-country comparisons, utilizing larger, more balanced samples and subgroup analyses to enhance the robustness of the findings. Moreover, while the CAMBRA protocol provides a useful framework for assessing caries risk, the exclusion of certain risk factors (*e.g.*, bacterial counts, salivary flow) due to logistical constraints may have limited the comprehensiveness of the risk assessment. Future research should aim to include these additional factors and explore longitudinal outcomes to better understand the effectiveness of CAMBRA in preventing caries progression in disabled populations.

## CONCLUSION

This study highlights significant disparities in oral health status, hygiene behaviors, and caries risk between individuals with disabilities in Pakistan and Saudi Arabia. Participants in both countries exhibited high caries risk, with differences in oral health outcomes and behaviors likely influenced by socioeconomic, cultural, and healthcare system factors. Saudi participants were found to have a slightly lower caries burden, potentially due to greater use of fluoridated toothpaste, yet their overall caries risk remained high, possibly

due to lifestyle factors such as frequent snacking. In Pakistan, deep pits and fissures were a predominant risk factor, indicating the need for preventive measures such as sealants and fluoride application. These findings highlight the critical need for tailored oral health education programs, improved access to preventive dental care, and specialized training for dental professionals to better serve disabled populations.

### Funding
This work was supported by the Deanship of Scientific Research at Jouf University through the fast-track research funding program. Ajman University, Ajman, UAE supported the APC of this article. The funders had no role in study design, data collection and analysis, decision to publish, or preparation of the manuscript.

### Grant Disclosures
The following grant information was disclosed by the authors:
Deanship of Scientific Research at Jouf University.
Ajman University, Ajman, UAE.

### Competing Interests
The authors declare there are no competing interests.

### Author Contributions
- Osama Khattak conceived and designed the experiments, authored or reviewed drafts of the article, and approved the final draft.
- Farooq Ahmad Chaudhary conceived and designed the experiments, authored or reviewed drafts of the article, and approved the final draft.
- Shahzad Ahmad performed the experiments, prepared figures and/or tables, and approved the final draft.
- Muhammad Amber Fareed conceived and designed the experiments, prepared figures and/or tables, and approved the final draft.
- Shazia Iqbal performed the experiments, prepared figures and/or tables, and approved the final draft.
- Asma Shakoor performed the experiments, prepared figures and/or tables, and approved the final draft.
- Mohammed Nadeem Baig performed the experiments, prepared figures and/or tables, and approved the final draft.
- Haifa Ali Almutairi analyzed the data, authored or reviewed drafts of the article, and approved the final draft.
- Rakhi Issrani analyzed the data, authored or reviewed drafts of the article, and approved the final draft.
- Azhar Iqbal analyzed the data, authored or reviewed drafts of the article, and approved the final draft.

## Ethics

The following information was supplied relating to ethical approvals (i.e., approving body and any reference numbers):

The ethical approval of this study was obtained from the ethical review board of Shaheed Zulfiqar Ali Bhutto Medical University (SZABMU) (Ref. No. SOD/ERB/2023/156).

## Data Availability

The data used for analysis of this study is available in the Supplementary File.

## Supplemental Information

Supplemental information for this article can be found online at http://dx.doi.org/10.7717/peerj.19286#supplemental-information.

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
