# Peer review of "Oral health status, oral hygiene behaviors, and caries risk assessment of individuals with special needs: a comparative study of Pakistan and Saudi Arabia"

_PeerJ, doi:10.7717/peerj.19286_

## Round 0.1 · original submission · Major Revisions

Dear Authors,

Please make sure to address all of the reviewer's comments thoroughly.

Reviewer 1 ·

Basic reporting

The article addresses a gap in the literature. This cross-country comparison adds value, considering the differences in healthcare access, social systems, and cultural attitudes toward oral health.

Experimental design

The methods are generally clear, but there could be additional detail in some areas like participant selection and statistical analysis.

Validity of the findings

The paper would benefit from a more detailed explanation of how it contributes to the literature. The conclusions are appropriately linked to the research question.

Additional comments

Please reformat some of your references to reflect the PeerJ References.

Example webpage:
"Johnson S. 2010. Italian Plants. Available at http://www.italianplants.com (accessed 22 March 2011)."

Reviewer 2 ·

Basic reporting

Strengths:
• The manuscript is written in clear, professional English with appropriate technical terminology
• The introduction provides good context and background on oral health challenges faced by disabled individuals
• Literature citations are relevant and current
• The structure follows standard scientific article format
• Tables are well-organized and clearly labeled
Areas for improvement:
• Some statistical results in the text are presented inconsistently with missing confidence intervals
• The methodology section would benefit from clearer description of calibration procedures and reliability measures
• Figures showing key findings would help visualize the results, particularly for DMFT scores and risk factors between countries

Experimental design

Strengths:
• Research questions and objectives are clearly defined
• Study fills an identified knowledge gap in comparing oral health status between two countries
• Methods for oral examination and risk assessment follow established protocols
• Ethical approval and consent procedures are well documented
Limitations requiring attention:
1. The sampling strategy needs more justification:
• Different margins of error used for the two countries limits comparability
• Convenience sampling may introduce selection bias
• Power calculations for cross-country comparisons are not provided
2. The calibration process requires more detail:
• Specific procedures for standardizing examiners
• Methods for calculating inter-examiner reliability
• Criteria for the Kappa value threshold
3. Exclusion criteria should be better justified:
• Rationale for age cut-off (13 years)
• Reasons for excluding certain disability types

Validity of the findings

Strengths:
• Appropriate statistical analyses used
• Raw data appears robust and comprehensive
• Conclusions are generally well-supported by results
Areas requiring attention:
1. Statistical analysis:
• Confidence intervals should be consistently reported
• Multiple testing corrections should be considered
• Effect sizes should be reported where relevant. Knowing the standardized effect size of oral hygiene interventions would help practitioners decide if implementing certain recommendations is worth the effort and resources
• Gingival index is an ordinary variable, so a Mann-Whitney U Test is more appropriate.
2. Potential confounders:
• Socioeconomic differences between countries need more discussion
• Impact of healthcare system differences on results
• Role of cultural factors in oral health behaviors
3. Limitations:
• The cross-sectional design limits causal inference
• Sample size disparities between countries affect comparability
• Generalizability concerns need more discussion

Additional comments

1. The study makes a valuable contribution by:
• Addressing an understudied population
• Providing comparative data between countries
• Using standardized assessment tools
2. Major improvements needed:
• Clearer justification for sampling strategy
• More detailed description of calibration procedures
• Better handling of potential confounders
• Addition of visual representations of key findings
3. Minor revisions suggested:
• More consistent reporting of statistical results
• Clearer organization of discussion section
• Addition of recommendations for future research
• Editing for conciseness in some sections
Recommendations
1. Major revisions recommended before acceptance:
• Address sampling strategy limitations
• Provide more detailed methodology
• Strengthen statistical reporting
• Add visual representations of results
2. Specific suggestions:
• Add flow diagram of participant selection
• Include forest plots for key comparisons
• Provide more detailed calibration procedures
• Discuss healthcare system differences between countries
This manuscript has merit but requires substantial revision before publication. The research question is important and the general approach is sound, but methodological details and analysis need strengthening.

·

Basic reporting

The manuscript needs improvement in clarity and consistency of the English language. Several grammatical errors and awkward phrasings require correction. Tables need reformatting for better readability. Literature citations are relevant but require updating and standardizing the reference format.

Experimental design

The study design is appropriate for the research question. However, the sample size calculation methodology needs revision, particularly regarding the different margins of error between countries (9% vs 4.65%). The calibration process for examiners requires a more detailed description. The modified CAMBRA protocol requires stronger justification for excluded components.

Validity of the findings

The statistical analysis is generally sound, though reporting needs standardization. Conclusions align with the results, but some causal inferences exceed the study's cross-sectional design limitations. The data appears robust and controlled, with appropriate statistical tests applied.

Additional comments

- Add power calculation for cross-country comparisons
- Expand limitations discussion
- Strengthen policy implications with evidence-based recommendations
- Standardize statistical reporting format
- Consider subgroup analyses by disability type
- Update self-citations (currently >15% of references)

---

## Round 0.2 · Major Revisions

Dear Authors,

Please address the comments and suggestions made by the reviewer. Thank you.

Reviewer 1 ·

Basic reporting

-Written in professional English with appropriate technical terminology. Some areas could benefit from minor revisions to improve clarity and readability.
-Additional visuals could enhance readability.

-Ensure consistency in statistical reporting

Experimental design

-Could elaborate on sampling could be improved

Validity of the findings

-Further elaboration on the impact of healthcare system differences would be beneficial.

·

Basic reporting

Strengths:
• The manuscript addresses an important research gap in comparing the oral health of individuals with disabilities across different healthcare systems.
• The authors have significantly improved the manuscript’s structure and clarity.
• The introduction effectively establishes the context and need for this study.

Areas for Improvement:

English Language and Grammar:
While notable improvements have been made, some grammatical issues remain throughout the manuscript. Please refer to the annotated PDF for further details.

Tables and Figures:
• The addition of figures showing DMFT values and caries risk assessment is helpful, but Figure 1 could benefit from clearer labeling of exact percentages within the pie charts.
• Table 3 still appears somewhat cluttered and could be reformatted for better readability, perhaps by grouping related questions together. Please refer to the annotated PDF for specific suggestions.

Experimental design

Strengths:
• The authors have provided much better justification for their methodological choices, particularly regarding the sample size calculations and margins of error.
• The detailed explanation of the calibration process significantly strengthens the methods section.
• The exclusion criteria are now well justified and explained.

Areas for Improvement:

Methodological Clarity:
• The authors justified using chi-square tests instead of Mann-Whitney U tests for analyzing Gingival Index data in their response to reviewers but did not include this rationale in the manuscript.

Statistical Analysis:
• The authors have added effect sizes, which strengthens the statistical reporting. However, they should clarify the practical significance of these effect sizes in the discussion section, particularly for the primary outcomes.
• The explanation for not performing subgroup analyses by disability type is reasonable but should be included in the limitations section of the manuscript.

Validity of the findings

Strengths:
• The inclusion of effect sizes enhances the interpretation of the results.
• The conclusions are generally aligned with the findings and research questions.
• The improved discussion of socioeconomic, cultural, and healthcare system differences strengthens the interpretation of the cross-country comparisons.

Areas for Improvement:

Results Interpretation:

• The contradictory finding that Saudi participants had lower DMFT scores but higher odds of being in the high-risk caries group still needs a more thorough explanation in the results or discussion section. To enhance clarity, consider adding a dedicated paragraph in the discussion section:

Suggested addition:
“An important finding requiring careful interpretation is the apparent contradiction between lower DMFT scores in Saudi participants coupled with their higher odds of being classified in the high-risk caries group. This seeming paradox may be explained by several factors. First, DMFT reflects past caries experience (a lagging indicator), while CAMBRA risk assessment evaluates current and future risk based on present behaviors and clinical indicators. The higher prevalence of risk factors such as frequent snacking (71.6%) and visible heavy plaque (60.4%) among Saudi participants suggests that despite their currently lower caries burden, they may be at greater risk for future caries development. Additionally, the different healthcare contexts may influence this pattern—Saudi Arabia's better access to dental care may have resulted in more restorative treatment (reflected in lower current DMFT scores) without necessarily addressing underlying risk behaviors. This finding highlights the importance of considering both past disease experience and current risk factors when developing preventive strategies.”

• Additionally, the relationship between oral hygiene behaviors and clinical findings should be explained more clearly, as this represents an interesting and potentially important observation. To address this, consider adding the following paragraph to the discussion section:

“Our findings reveal an interesting disconnect between self-reported oral hygiene behaviors and clinical outcomes. Despite Saudi participants reporting significantly higher twice-daily brushing rates (34.6% vs. 13.0% in Pakistan), they exhibited similar rates of visible plaque (35.8% vs. 34.3%) and gingival inflammation. This disconnect suggests that either the quality of brushing is suboptimal, reporting bias exists in self-reported behaviors, or other factors such as diet and genetic predisposition play more significant roles in determining clinical outcomes than brushing frequency alone. This observation underscores the need for comprehensive oral health education emphasizing proper brushing technique and duration, not just frequency, particularly for individuals with disabilities who may have additional challenges in performing effective oral hygiene.”

Discussion:

• While improved, the discussion section could be further strengthened by organizing it more clearly around the main findings before delving into implications. Below is a suggested restructuring format:
1. Brief summary paragraph highlighting key findings
2. Dedicated paragraphs for each major finding:
- DMFT differences between countries
- Gingival health and visible plaque findings
- Oral hygiene behavior patterns and discrepancies
- Caries risk assessment findings and contradictions
- Protective factors identified
3. Interpretation of findings in the context of healthcare systems and cultural differences
4. Implications for practice and policy
5. Strengths and limitations
6. Directions for future research
Implementing this structure would enhance readability and ensure that key findings and their implications are clearly conveyed. Please consider this approach to improve the overall coherence of the discussion section.

• The policy recommendations lack specificity to each country’s context based on the unique findings.

Additional comments

Dear Authors,
Thank you for submitting your revised manuscript and for thoroughly responding to the previous reviewers’ comments. I have carefully evaluated the revised version in light of these revisions and appreciate the improvements made. Above is my updated assessment.

Furthermore, the conclusion could be made more impactful by emphasizing the most significant and unexpected findings, along with their implications for future research and clinical practice.

With these refinements, I believe the study would enhance understanding of oral health disparities in individuals with disabilities and guide targeted interventions to improve their oral health outcomes.

---

## Round 0.3 · accepted · Accept

Dear authors, thank you for considering the reviewer's suggestions.